# Optimizing the Choice for Adjuvant Chemotherapy in Gastric Cancer

**DOI:** 10.3390/cancers14194670

**Published:** 2022-09-25

**Authors:** Antonino Grassadonia, Antonella De Luca, Erminia Carletti, Patrizia Vici, Francesca Sofia Di Lisa, Lorena Filomeno, Giuseppe Cicero, Laura De Lellis, Serena Veschi, Rosalba Florio, Davide Brocco, Saverio Alberti, Alessandro Cama, Nicola Tinari

**Affiliations:** 1Department of Innovative Technologies in Medicine and Dentistry, and Center for Advanced Studies and Technology (CAST), G. D’Annunzio University Chieti-Pescara, 66100 Chieti, Italy; 2Department of Medical, Oral and Biotechnological Sciences, and Center for Advanced Studies and Technology (CAST), G. D’Annunzio University Chieti-Pescara, 66100 Chieti, Italy; 3Unit of Phase IV Trials, IRCCS Regina Elena National Cancer Institute, 00144 Rome, Italy; 4Department of Surgical, Oncological and Oral Sciences, Section of Medical Oncology, University of Palermo, 90133 Palermo, Italy; 5Department of Pharmacy, G. D’Annunzio University Chieti-Pescara, 66100 Chieti, Italy; 6Unit of Medical Genetics, Department of Biomedical Sciences—BIOMORF, University of Messina, 98125 Messina, Italy

**Keywords:** gastric cancer, adjuvant chemotherapy, predictive factors, prognostic factors

## Abstract

**Simple Summary:**

Gastric cancer is the fourth largest cause of tumor-related death worldwide. Despite advances in the management of resectable cancer and improvements in early diagnosis, especially in east Asia where screening campaigns are actively performed, many patients experience recurrence and die because of the disease. Adjuvant systemic chemotherapy is administered after radical surgery in order to reduce the risk of recurrence and death. The modality of administration and regimens of chemotherapy in this setting are different between Eastern and Western countries. In Asia, adjuvant chemotherapy is traditionally given after surgery, while in Europe it is commonly scheduled after preoperative chemotherapy and surgery (perioperative chemotherapy), and in Northern America it is usually combined with radiotherapy (chemoradiotherapy). All these approaches are sustained by well-designed phase III clinical studies, and none may be considered superior to the others in the absence of head-to-head comparisons. The identification of predictive and/or prognostic factors could help to select patients at higher risk of recurrence and those more likely to receive a benefit from the adjuvant treatment. This would allow clinicians to avoid the administration of undue toxicity to non-responder patients and even to reduce the cost of unnecessary treatment.

**Abstract:**

Advances in the management of gastric cancer have improved patient survival in the last decade. Nonetheless, the number of patients relapsing and dying after a diagnosis of localized gastric cancer is still too high, even in early stages (10% in stage I). Adjuvant systemic chemotherapy has been proven to significantly improve outcomes. In the present article we have critically reviewed the clinical trials that guide the current clinical practice in the adjuvant treatment of patients affected by resectable gastric cancer, focusing on the different approaches worldwide, i.e., adjuvant chemotherapy, adjuvant chemoradiotherapy, and perioperative chemotherapy. We also delineate the clinical–pathological characteristics that are commonly taken into account to identify patients at a higher risk of recurrence and requiring adjuvant chemotherapy, and also describe novel biomarkers and therapeutic agents that might allow personalization of the treatment.

## 1. Introduction

The incidence of gastric cancer has been constantly decreasing over the past decades in Western countries, while remained particularly high in Eastern Asia. In 2020, gastric cancer has been reported as the 16th more diagnosed cancer in the United States [1], but accounted for the first more common cancer in Japan and Korea [2]. Given the high mortality-to-incidence ratio, gastric cancer is the forth cause of tumor-related death worldwide, after lung, colorectal, and liver cancer [3].

Radical surgery might potentially be curative for patients with resectable gastric cancer, but unfortunately many of them experience loco-regional and/or metastatic recurrence and die because of the disease. In some series the recurrence rate can reach 70% with a 5-year overall survival (OS) rate of 20–30% in Western countries. Among recurring patients, 75% relapse within the first 2 years after the completion of local therapy and almost all (90–95%) within 5 years.

Several studies have provided evidence that additional treatment to surgery can significantly reduce recurrence and improve survival, especially in patients with stage > II. Different therapeutic approaches have been utilized worldwide. In Asia, chemotherapy is administrated after radical surgery, i.e., as adjuvant chemotherapy, while in USA, adjuvant chemoradiotherapy is preferred. In Europe, chemotherapy is administrated in part before surgery (neoadjuvant chemotherapy) and in part after surgery (adjuvant chemotherapy), strategy referred to as perioperative chemotherapy.

Here we report a critical review of the clinical trials that support the current management of resectable gastric cancer and analyze the clinical–pathological factors associated with higher risk of recurrence, with the aim to provide clinicians with a practical tool that might help in the identification of patients who are more likely to benefit from adjuvant chemotherapy. In addition, we discuss the reasons underlying the differences between Western and Eastern approaches. Finally, a brief description of the emerging predictive biomarkers and of the novel therapeutic agents that could potentially impact on clinical practice in the near future is also included.

## 2. Factors Influencing Recurrence after Radical Surgery

### 2.1. Tumor Stage

The most relevant prognostic factor for survival remains stage, currently based on the 8th edition of the AJCC TNM staging system [4]. Outcome data for this last edition come from more than 25,000 patients with gastric cancer who had undergone radical surgery with adequate lymph node dissection and was included in the International Gastric Cancer Association (IGCA) database. The 5-year OS rate was 92% for stage IA, 88% for stage IB, 81% for stage IIA, 68% for stage IIB, 54% for stage IIIA, 36% for stage IIIB, and 18% for stage IIIC. Most of these patients were from Japan and Korea (84.8%), but data have been validated in populations in the USA [5,6] and China [7], as well.

Nearly 70–80% of patients diagnosed with localized gastric cancer have involvement of the regional lymph nodes, and the number of positive lymph nodes, along with tumor size, has a profound influence on survival [8]. In a retrospective study, 7371 surgically treated patients selected from the Surveillance, Epidemiology, and End Results (SEER) program with at least 16 lymph node removed were staged according to the 8th edition of TNM classification in order to verify its ability to predict long-term outcomes [9]. The TNM staging system demonstrated the capacity to finely discriminate survival among different stages. In particular, significant differences were observed among patients with the same tumor size (pT), but with a different number of positive lymph nodes (pN). For example, the 5-year OS rate was 90.6% for patients with a pT1 primary tumor, i.e., solely infiltrating mucosa and submucosa, but it progressively worsened with increasing number of positive lymph nodes: 75.4% in pN1 (1–2 nodes positive), 74.4% in pN2 (3–6 nodes positive), 66.8% in pN3a (7–15 nodes positive), 37.5% in pN3b (>15 nodes positive) [9].

It is then evident that the choice to administer adjuvant chemotherapy is strictly dependent on the patient’s prognosis, as defined by TNM classification. Moreover, in order to avoid misclassification, at least 16 lymph nodes should be removed and examined by pathologist, as discussed below.

### 2.2. Number of Examined Lymph Nodes

Inadequate lymphadenectomy may be responsible for understaging gastric cancer and can affect long-term outcomes. In a cohort of 4670 patients with pN0 gastric cancer (66% stage I) selected from the SEER database, stratified according to the retrieved and examined lymph node count (1–4 vs. 5–13 vs. >13 lymph nodes), the 5-year cause-specific survival rate was significantly improved by increasing lymph nodes retrieval, 64.8%, 72.5%, and 79.4%, respectively (5–13, HR 0.684, 95% CI 0.589–0.794, *p* < 0.001; > 13, HR 0.501, 95% CI 0.428–0.587, *p* < 0.001; 1–4 as reference) [10]. A similar retrospective analysis of SEER database found that in 1971 patients with stage IB gastric cancer who underwent radical surgery, number of examined lymph nodes ≤15 and T1N1M0 stage were independent predictors of worse OS [11]. Another retrospective study analyzed patients with pT2N0 gastric cancer from the National Cancer Data Base and reported that the 5-year OS rate was 71% for patients with ≥15 lymph nodes vs. 53% for those with <15 examined lymph nodes (HR 0.61, *p* < 0.001) [12]. In addition, in the group of patients with a lymph node count <15, those treated with chemoradiotherapy showed a better OS compared to those who had surgery alone (HR 0.71, 95% CI 0.50–0.99, *p* = 0.043) [12].

Based on the above evidence, the examination of less than 16 lymph nodes is considered a risk factor of disease recurrence, which would suggest the use of adjuvant treatment even for early-stage cancers, when an insufficient number of nodes is analyzed. Notably, it has been estimated that in the USA half of the patients with gastric cancer had fewer than 16 lymph nodes examined [13].

### 2.3. Type of Lymphadenectomy

Lymph node dissection is classified as D1, when perigastric lymph nodes are removed, or D2, when the removal of all the lymph nodes along the left gastric artery, common hepatic artery, celiac artery, and splenic artery is added to D1 dissection. Modified radical lymphadenectomy, less extensive than D2 but more extensive than D1, is also performed in clinical practice, e.g., D1.5 or D1+, when D1 lymphadenectomy is extended with dissection along the left gastric and common hepatic arteries.

Eastern surgeons routinely perform the more radical lymphadenectomy D2, and this practice has been recommended by the Japanese guidelines since 1981. In Western countries, the use of D2 dissection is more limited on the basis of two randomized studies, the UK Medical Research Council Trial and the Dutch Gastric Cancer Trial, both of which failed to demonstrate a survival benefit for D2 over D1, while reporting a higher postoperative morbidity and mortality associated with D2 surgery [14,15]. However, major criticisms have been raised about these studies, such as unacceptably high mortality, contamination (i.e., lymph nodes removed outside the intended level of resection), and high rate of non-compliance (i.e., inadequate removal of lymph node stations) [16]. A revision of the Dutch trial revealed a major non-compliance in 26% of the D2-resected patients and, after exclusion of these patients, the 15-year OS rate was significantly better in D2 than in D1 group, 35.7% and 19.9%, respectively (*p* = 0.041) [17].

Nowadays, D2 surgery is considered the standard of surgical treatment for medically fit patients, and less-than-D2 lymph node dissection is considered a risk factor of recurrence. However, to reduce the rate of non-compliance and the risk associated with the procedure, D2 lymphadenectomy should be performed in high-volume centers by experienced surgeons.

### 2.4. Tumor Grade

Two retrospective studies analyzed the risk of recurrence associated with tumor grade in patients with stage I disease. The first study assessed 2783 surgically treated patients and revealed that poor tumor differentiation, mostly observed in cancer with diffuse histology, compared to good or moderate tumor differentiation, was associated with reduced recurrence-free survival (RFS) (*p* = 0.001) [18]. The other study analyzed 86 patients and showed that those with histologically undifferentiated gastric adenocarcinoma (G3, high grade) were at higher risk of recurrence (*p* = 0.0069) [19]. The overall 5-year survival rate of patients with undifferentiated-type adenocarcinoma was lower than that of the patients with differentiated-type (84% vs. 100%) [19].

Therefore, even in the absence of randomized controlled studies, it is conceivable that poorly differentiated early-stage gastric cancer might benefit from adjuvant chemotherapy.

### 2.5. Lymphovascular and/or Perineural Invasion

Metastatic progression of gastric cancer has been associated with the presence of tumor lymphovascular and/or perineural invasion [20]. A retrospective analysis of 225 patients with stage I gastric cancer who had received R0 resection, reported that lymphovascular and perineural invasion were significantly associated with reduced OS (HR 3.85, 95% CI 1.75–8.48, *p* = 0.001 and HR 4.03, 95% CI 1.84–8.83, *p* < 0.001, respectively) and that adjuvant chemotherapy was able to improve outcome in patients with these tumor characteristics (HR 0.447, 95% CI 0.447–0.996, *p* = 0.042) [21]. Another retrospective study analyzed 130 patients with stage IB node-negative gastric cancer (pT2pN0) and found that tumor venous invasion was independently predictive of reduced RFS (HR 3.00, 95% CI 1.08–9.59, *p* = 0.035) and OS (HR 5.00, 95% CI 1.54–22.36, *p* = 0.006) [22].

In a prospective study of 233 patients with stage I gastric cancer (pT2pN0), who had undergone D2 lymph node dissection, the 5-year OS rate was lower in patients affected by tumor with lymphovascular or perineural invasion (HR 3.09, 95% CI 0.50–5.01, *p* = 0.025 and HR 4.83, 95% CI 0.60–6.88, *p* = 0.009, respectively) compared with patients without these tumor characteristics [23].

In summary, although the presence of lymphovascular/perineural invasion has not been formally analyzed in relation to different stages of gastric cancer, it appears to predict a poor outcome, and as such it should be considered when deciding for adjuvant chemotherapy.

### 2.6. Age of Patient

A retrospectively analyses of 597 patients who underwent D2 gastrectomy for early-stage gastric cancer was performed in order to establish the factors associated with the likelihood of tumor lymph node involvement. Age < 50 years, along with lymphovascular invasion and poor differentiation, emerged as independent predictors of lymph node metastasis (RR 0.444, 95% CI 0.215–0.916, *p* = 0.028) [24].

Thus, age <50 years should be considered among the factors favoring the use of adjuvant chemotherapy.

### 2.7. Immunological and Nutritional Status of Patient

Parameters related to the host inflammatory/immune response, such as C-reactive protein (CRP), Interleukin-6 (IL-6), lymphocyte-to-CRP ratio (LCR), neutrophil–lymphocyte ratio (NLR), lymphocyte–monocyte ratio (LMR), platelet–lymphocyte ratio (PLR), and immune inflammation index (SII) [25,26,27], and factors associated with the nutritional status of patients, including serum albumin, prealbumin (PA), body mass index (BMI), skeletal muscle area (SMA), skeletal muscle index (SMI), malnutrition screening tool (MST), Naples Prognostic Score (NPS), prognostic nutritional index (PNI), and nutritional risk index (NRI) [28,29,30], have extensively been suggested as predictive indicators for both prognosis and postoperative complications in patients with gastric cancer. A recent retrospective study, carried out in a cohort of patients with stage II–III gastric cancer who survived longer than 1 year after curative gastrectomy and adjuvant chemotherapy, revealed that a preoperative good nutritional status (high NRI) or a minimal skeletal muscle loss (lower difference in SMI, dSMI) were significantly associated with better prognosis. The 5-year OS rates were 75.8%% vs. 63% in patients with high NRI and low NRI, respectively (HR 0.96, 95% CI 0.948–0.973, *p* < 0.001), and 75.7% vs. 66.2% in patients with low dSMI and high dSMI, respectively (HR 0.95, 95% CI 0.938–0.981, *p* < 0.001) [31].

However, the prognostic significance of immuno-nutritional indices in patients with different disease stages remains unclear and we have no evidence that supporting nutrition and preserving muscle mass can improve long-term outcomes. The phase III clinical trials that guide the current clinical practice (discussed below) did not consider these parameters, and enrolled fit patients with good performance status.

## 3. Trials Supporting Current Clinical Practice

Several phase III clinical trials have established that the high risk of recurrence and death observed with surgery alone can be significantly reduced by the addition of systemic chemotherapy or chemoradiotherapy. The timing and the modality of this strategy is different in Western and Eastern countries, mostly driven by the results of the phase III clinical trials carried out in the respective countries.

Landmark trials, along with the country of the studied population, are reported in Table 1, grouped in (i) adjuvant chemotherapy, (ii) adjuvant chemoradiotherapy, and (iii) perioperative chemotherapy.

### 3.1. Adjuvant Chemotherapy

Three phase III trials, all performed among Asian populations, demonstrated the efficacy of adjuvant chemotherapy.

The Japanese Adjuvant Chemotherapy Trial of S-1 for Gastric Cancer (ACTS-GC) enrolled 1069 patients who had undergone D2 surgery for stage II or III gastric adenocarcinoma [32]. Patients were randomized to the oral fluoropyrimidine S-1 for 12 months or observation. The trial was stopped early when the S-1 group showed a higher OS rate compared to the surgery-only group (*p* = 0.002). Results were updated at longer follow-up and showed a 10% increase in 5-year OS in the S-1 group compared to the observation group (71.7% vs. 61.1%, HR 0.669, 95% CI 0.540–0.828) [33]. The RFS rates at 5 years were 65.4% and 53.1% in the S-1 group and surgery-only group, respectively (HR, 0.653; 95% CI, 0.537 to 0.793). Based on this trial, adjuvant S-1 became the standard practice in East Asia.

The JACCRO GC-07 (START-2) trial was carried out exclusively on patients with stage III disease after D2 resection, and compared outcome of patients who received single-agent S-1 for 12 months (N = 459) with those who received S-1 plus docetaxel for six cycles (N = 453) [34]. The 3-year RFS rate of the combination group was significantly superior to that of the S-1 group, 67.7% and 57.4%, respectively (HR 0.715, 95% CI 0.587–0.871, *p* = 0.0008). A significant benefit for the combination vs. the S-1 group was observed also in the 3-year OS rate (77.7% and 71.2%, respectively; HR 0.742, 95% CI 0.596–0.925, *p* = 0.0076) [34].

The CLASSIC trial evaluated the efficacy of adjuvant chemotherapy with capecitabine and oxaliplatin (CAPOX) after D2 lymph node dissection in 1035 patients with stage II-IIIB gastric cancer [35]. Patients were randomized to receive either surgery alone (N = 515) or surgery followed by postoperative chemotherapy (N = 520). Independently from the disease stage, patients who received CAPOX showed a significantly improved 3-year disease-free survival (DFS) rate compared to those treated with surgery alone (74% and 59%, respectively; *p* < 0.0001). The study, updated after a median follow-up of 62.4 months, showed an improved 5-year DFS rate (68% vs. 53%) and OS rate (78% vs. 69%) in the adjuvant chemotherapy group as compared to the option of surgery alone [36].

Overall, adjuvant studies have clearly established a role for chemotherapy in increasing survival in patients who have undergone surgery for resectable gastric cancer. Even if a direct comparison is not practicable, doublet chemotherapy (capecitabine plus oxaliplatin) seems to achieve a better clinical benefit compared to single-agent therapy (S-1). None of these studies enrolled patients at stage I, thus it is not possible to determine if patients with limited disease can receive the same clinical benefit from adjuvant chemotherapy. The merit of enrolling patients with appropriate D2 lymphadenectomy, routinely performed in Asia, might be a limit for the extension of these results in Western countries where D1 or D1+ dissection is usually performed. In addition, the oral fluoropyrimidine S-1 agent has not been approved thus far in the USA and in some European countries.

### 3.2. Adjuvant Chemoradiotherapy

The intergroup-0116 (INT-0116) trial, carried out more than 20 years ago in North America, showed for the first time the possibility of improving outcomes of patients with completely resected gastric cancer using adjuvant chemotherapy plus chemoradiotherapy [37]. The trial randomized 556 surgically treated gastric cancer patients, mostly with pT2-pT3 (69%) and lymph node positive (85%) tumors, into groups undergoing observation or chemotherapy with 5-fluorouracil (5-FU)/folinic acid (FA) in bolus on days 1–5 q28 for 5 cycles with the addition of radiotherapy 45 Gy in 25 fractions over 5 weeks (5 days per week) starting from day 1 of cycle 2 up to day 5 of cycle 3. Median OS was 36 months in the chemoradiotherapy arm and 27 months in the surgery-alone arm (HR 1.35, 95% CI 1.09–1.66; *p* = 0.005), while median RFS was 30 months and 19 months in the two arms, respectively (HR 1.52, 95% CI 1.23–1.86; *p* < 0.001).

The Korean ARTIST trial evaluated the addition of radiotherapy to chemotherapy in 458 patients treated with surgery and D2 dissection. Patients were randomized to six cycles of capecitabine plus cisplatin (XP) or XP plus radiotherapy and capecitabine (XPRT) [38,39]. The addition of radiotherapy to XP chemotherapy did not significantly improve the 3-year DFS rate (78.2% vs. 74.2%, *p* = 0.0862). However, in a subgroup analyses, among patients with pathologic lymph node metastasis (N = 396), those treated with chemoradiotherapy had a better DFS (*p* = 0.0365).

The ARTIST II trial was a 3-arm study designed to verify the superiority of adding oxaliplatin to S-1, and whether the addition of radiotherapy to this regimen would improve survival in patients who were node-positive after D2 dissection [40]. Patients (N = 546) were randomized in three adjuvant regimens: oral S-1 for 1 year, S-1 plus oxaliplatin (SOX) for 6 months, and SOX plus chemoradiotherapy 45 Gy (SOXRT). The 3-year DFS rates were 64.8%, 74.3%, and 72.8%, respectively, with a significant improvement in the SOX arm compared to the control one (S-1) (HR 0.69, *p* = 0.042). On the contrary, no difference in DFS was found between SOX and SOXRT (HR 0.971; *p* = 0.879), indicating no benefit from RT when D2 dissection is performed.

Overall, adjuvant chemoradiotherapy studies indicate that the addition of radiotherapy to chemotherapy can be omitted in the presence of D2 lymphadenectomy. The ARTIST II trial also established that in stage III gastric cancer, a doublet chemotherapy (S-1 plus oxaliplatin) is superior to single-agent therapy (S-1). The limit of the INT-0116 trial was inadequate lymphadenectomy, with only 10% of patients undergoing D2 resection. Thus, although an updated analysis of the study confirmed a better OS for patients treated with adjuvant chemoradiotherapy (HR 1.32, 95% CI 1.10–1.60; *p* = 0.0046) [41], the benefit of this approach could be justified by the overall higher risk of relapse in the studied population [42,43].

### 3.3. Perioperative Chemotherapy

In Europe, clinical practice in the adjuvant setting has been largely influenced by two trials of perioperative chemotherapy, MAGIC and FLOT-4.

The MAGIC trial was the first pioneering study that randomized 503 patients with resectable gastric (74%), gastro-oesophageal junction (14.5%) and lower oesophageal (11.5%) cancer to receive perioperative chemotherapy with epirubicin, cisplatin and 5-FU (ECF) (3 cycles before and 3 cycles after surgery), or surgery alone [44]. Patients who were treated with perioperative chemotherapy showed a significantly improved 5-year OS rate compared to those treated with surgery alone (36% and 23%, respectively; HR 0.75, 95% CI 0.60–0.93; *p* = 0.009). Afterward, several other studies demonstrated the efficacy of perioperative chemotherapy in resectable gastric cancer [45].

More recently, the FLOT4 trial has reported that the combination of 5-FU, FA, oxaliplatin and docetaxel (FLOT) was superior to ECF or ECX (capecitabine instead of 5-FU) regimens [46]. In this study 716 patients with operable gastric cancer were randomly assigned to receive perioperative FLOT (4 cycles before and 4 cycles after surgery) or ECF(X) (3 cycles before and 3 cycles after surgery). Patients treated with the FLOT triplet showed a significantly better survival rate compared to those treated with ECF(X). The 5-year OS rates were 45% and 36% for the two regimens, respectively (HR 0.77; CI, 0.63–0.94; *p* = 0.012) [46].

The limit of the perioperative approach is the risk of misestimating the stage of gastric cancer since it is based on imaging procedures (clinical staging) that can overstage or understage the disease. However, perioperative chemotherapy is endowed with some advantages: (i) greater likelihood to deliver the full dose of systemic therapy, given that patient population usually retains a good performance status; (ii) neoadjuvant chemotherapy can downstage locally advanced tumors, allowing a more radical surgery; (iii) improve the identification of patients for whom surgery may not offer a survival benefit because of disease progression during the treatment.

It is important to note that the results of perioperative chemotherapy were obtained with only 54.8% of patients in the MAGIC trial and 60% of patients in the FLOT4 trial completing the post-operative part of the treatment. Thus, it can be hypothesized that some patients responding to preoperative chemotherapy would be spared an additional adjuvant treatment. This is the objective of an ongoing clinical trial in which perioperative chemotherapy with epirubicin, oxaliplatin and capecitabine (EOX) (3 cycles before and 3 cycles after surgery) is compared with neoadjuvant EOX (just 3 cycles before surgery) (NCT01787539, STOPEROPCHEM).

## 4. Current Indications for Adjuvant Chemotherapy

### 4.1. Stage II–III

Given the high risk of recurrence after surgery alone and based on the above-described phase III clinical trials, there is a complete consensus among oncologists to recommend adjunctive therapy to curative surgery in patients with stage II–III gastric cancer, either adjuvant or perioperative, depending on the country of treatment.

In East Asia, adjuvant chemotherapy, including single agent S-1, doublet S-1 plus docetaxel, or capecitabine plus oxaliplatin, is recommended by the Japanese gastric cancer treatment guidelines (JGCG) [47].

In USA, adjuvant chemoradiotherapy with 5-FU/FA is the standard treatment according to the National Comprehensive Cancer Network (NCCN) guidelines [48]. However, considering the high toxicity of the original regimen associated with radiotherapy (5-day bolus of 5-FU/FA every 28 days), modifications based on oral capecitabine [49] or 5-FU according to the De Gramont schedule [50] are encouraged. In case of D2 dissection, adjuvant chemotherapy without radiotherapy can be considered.

In Europe, perioperative chemotherapy with the FLOT regimen is now considered the standard of care by the European Society of Medical Oncology (ESMO) guidelines [51]. Adjuvant chemotherapy or chemoradiotherapy are to be considered in case of patients who have undergone upfront surgery without administration of preoperative chemotherapy.

All these approaches are able to improve survival compared to surgery alone in stage II–III gastric cancer, but randomized phase III clinical trials to establish if one is better than the others are lacking. A recently published study, the RESOLVE trial, carried out on selected patients with locally advanced gastric or gastro-oesophageal junction cancer (cT4a N+ M0 or cT4b Nany M0) undergoing D2 gastrectomy, showed that SOX was non-inferior to CapOx in the adjuvant setting, but that perioperative SOX was associated with better DFS compared to adjuvant CapOx. The 3-year DFS rate was 59.4% in the perioperative–SOX group compared to 51.1% in the CapOx–adjuvant group (HR 0.77, 95% CI 0·61–0·97; *p* = 0.028) [52].

### 4.2. Stage I

Despite the general good prognosis of stage I gastric cancer, it recurs in about 10% of patients within 5 years. While there is a general consensus for surveillance alone in patients with stage IA (pT1N0) tumors, controversies exist regarding the more appropriate treatment of patients at stage IB, which encompasses pT1N1 and pT2N0 tumors according to the 8th edition of the AJCC TNM classification [4].

Two opposite approaches are endorsed by JGCG and ESMO organizations in their respective guidelines for the treatment of stage IB gastric cancer. While the former does not recommend adjuvant therapy (only observation), the latter recommends perioperative chemotherapy as preferred option, exactly as indicated for stage II and III tumor. It is clear that the decision of perioperative chemotherapy is taken on the basis of a clinical stage defined by imaging (CT scan or endoscopic ultrasound), with the risk of tumor understaging. Thus, in case of patients clinically evaluated at stage IA who have undergone surgery without administration of preoperative chemotherapy, ESMO recommends adjuvant chemotherapy or chemoradiotherapy if the definitive pathological stage results ≥ IB. NCCN makes a distinction between patients with pT1N1 and those with the pT2N0 stage. For patients diagnosed at the pT1N1 stage, chemotherapy or chemoradiotherapy (if less than D2 dissection) are recommended, while for those with pT2N0 stage follow-up without adjuvant treatment can be appropriate, unless high-risk tumor characteristics are present such as age ≤ 50 years, poorly differentiated/high-grade cancer, lymphovascular invasion, neural invasion or less-than-D2 lymph node dissection. In these cases, adjuvant chemoradiotherapy is recommended.

At the moment, no randomized trial has been specifically focused on the adjuvant treatment of patients with stage IB disease. On the other hand, very few of these patients have been included in the landmark clinical trials, as shown in Table 2. Therefore, improvement of survival observed in these trials cannot be assumed for patients with stage IB. Insights can be gained from retrospective studies.

A recent analysis of the SEER database investigated the impact of adjuvant chemotherapy in 1767 patients diagnosed with stage IB gastric cancer [53]. Patients who received chemotherapy showed a 5-year cumulative incidence of cancer-specific death, significantly lower than those in the no-chemotherapy group (11.5% vs. 20.8%, *p* = 0.007). Another retrospective SEER-based analysis unexpectedly found that stage I patients (N = 8335) received more benefit from surgery alone compared to adjuvant chemotherapy. However, this finding was affected by the high number of stage IA patients (30% of the studied population) that would probably not have received benefit from adjuvant chemotherapy. Indeed, when stage IA patients were excluded, the remaining stage IB patients treated with chemoradiotherapy showed a better survival as compared to those treated with surgery alone (HR, 1.29; CI, 1.14–1.46) [54]. Thus, the result of this study indicates that adjuvant therapy can improve survival in patients with stage IB, and that additional treatment is not necessary in patients with stage IA.

Similar results were observed in a study of 9947 patients with stage IB gastric cancer selected from the National Cancer Database [55]. Patients treated with adjuvant chemotherapy or chemoradiotherapy showed a significantly lower risk of death compared to surgery alone (HR 1.23 CI 1.09–1.39, *p* = 0.001). However, in patients with node-negative disease, the benefit from the adjuvant treatment was lost in the presence of an adequate lymph node dissection (≥16), indicating that patients with stage pT2N0 could avoid adjuvant therapy when adequate lymphadenectomy is performed [55]. This aspect was also investigated in a SEER database cohort of 1971 patients with stage IB gastric cancer where adjuvant chemotherapy or chemoradiotherapy significantly improved survival of patients with the pT1N1 stage, but not of those with pT2N0 [11]. The study also showed that pT2N0 had a significantly better survival rate than pT1N1, but this advantage was lost when restricting the analysis to patients with ≥16 dissected lymph nodes, indicating that extended lymphadenectomy can improve survival in pT1N1 disease as well.

Although included in the same stage group according to the 8th edition of TNM classification, pT2N0 appears to have a better prognosis than pT1N1. The difference is more evident in the case of <16 dissected lymph nodes [11], but it is still appreciable when adequate lymphadenectomy is performed. In fact, a SEER cohort of 7371 patients who had undergone surgery with at least 16 lymph node removed, showed a 5-year OS rate of 85.9% in pT2N0 and 75.4% in pT1N1 stage, i.e., a more-than-10% increase in survival for the pT2N0 group [9].

In the absence of randomized clinical trials focusing on stage IB gastric cancer, routine adjuvant chemotherapy cannot be recommended. Overall, the retrospective studies emphasize a possible benefit from adjuvant chemotherapy in patients with pT1N1 gastric cancer. Patients staged as pT2N0 have a better prognosis and might potentially receive a lower benefit from adjuvant treatment. Nonetheless, as described above, some clinical–pathological characteristics of pT2N0 tumors can be associated with an increased risk of recurrence [56], suggesting that adjuvant chemotherapy might be considered.

## 5. Novel Biomarkers and Therapeutic Approaches

Intensive experimental research is currently focused on biomarkers that are potentially relevant for the selection of patients who may benefit from additional treatment beside surgery, including not only the conventional chemotherapeutic agents, but also the emerging immune checkpoint inhibitors (ICI) and molecularly targeted therapies. The identification of different molecular subtypes of gastric cancer has greatly accelerated this process.

### 5.1. Molecular and Microenvironmental Diversity of Gastric Cancer

In 2014, The Cancer Genome Atlas (TCGA) Consortium performed molecular profiling of gastric cancer in a cohort of patients not treated with prior chemotherapy or radiotherapy [57]. By comparing the genomic/molecular pattern in tumor and normal tissues, four distinct cancer subtypes were identified: (i) Epstein–Barr virus (EBV)-positive tumors, that show DNA hypermethylation, PIK3CA mutations, and PDL-1/2 amplification; (ii) tumors with microsatellite instability (MSI), showing high tumor mutational burden; (iii) genomically stable tumors (GS), characterized by an activated function of the Rho family of GTPases proteins; and (iv) tumors with chromosomal instability (CIN), which show aneuploidy and amplification of receptor tyrosine kinases [57].

Using TCGA classification in a cohort of patients who had undergone surgery for gastric cancer, a prediction model for response to adjuvant chemotherapy and survival has been developed [58]. Specifically, EBV subtype appears to be associated with the best prognosis, MSI and CIN with a moderate prognosis, while GS showed the worst one. When analyzing the benefit from adjuvant chemotherapy, patients with CIN subtype showed the greatest benefit, while those with GS received no benefit, and only a moderate benefit was observed among patients with MSI subtype. Since all patients with the EBV subtype received adjuvant chemotherapy, benefit could not be assessed in this subgroup [58].

A great heterogeneity in the immune cell infiltration of tumor microenvironment has been observed across the different TGCA subtypes. EBV and MSI tumors have intense T-cell infiltrates, high PD-L1 expression [59], and exhibit better response to ICI [60]. The immune responsiveness of EBV subtype has been attributed to the expression of viral antigens by cancer cells, while that of MSI is likely due to the high rate of mutations that promote neo-antigens formation. On the contrary, CIN and GS tumor showed reduced T-cell infiltrates [59,61] and poor responses to ICI [60]. Although the reasons of immune suppression in CIN and GS are not known, a mechanism of immune exclusion has been proposed, at least for CIN, where CD 8+ T-cells seem to be incapable of infiltrating tumor microenvironments and cluster at the invasive margin of the neoplastic tissue [62].

A recent study has depicted the molecular profile of the tumor-infiltrating immune cells [63]. Based on the expression of 730 immune-related genes, an immunogenomic characterization score (IGCS) has been developed and tumors were classified as IGCS-low when abundant innate or adaptive immune cell infiltration was present. IGCS-low was associated with enhanced responses to anti-checkpoint inhibitors in four different immunotherapy cohorts [63]. Interestingly, determination of the IGCS in the different TCGA molecular subtypes showed that higher IGCS was expressed in CIN and GS, while lower IGCS was present in MSI and EBV [63], indicating that the latter were more immunogenic and responsive to ICI.

Although gene profiling of gastric cancer is not applicable in clinical practice due to the objective difficulty of performing routine high throughput analyses, it clearly revealed the existence of different molecular subtypes of gastric cancer that display unique biological behavior and different responsiveness to systemic treatment. Techniques available in routine diagnostic practice, including immunohistochemistry (IHC), polymerase chain reaction (PCR), and in situ hybridization (ISH), can easily allow researchers evaluate specific markers to be utilized as surrogate for the molecular subtypes. At the moment, the assessment of MSI in cancer cells, the detection of EBV-infected cancer cells, and the expression of PDL-1 in cancer cells or in the immune-infiltrating cells are the emerging markers that are likely to change the adjuvant/perioperative approach of gastric cancer, as detailed below.

### 5.2. Microsatellite Instability (MSI)

MSI is the result of defects in the mismatch repair system (MMR), a mechanism responsible for repairing incorrect matching between nucleotides during normal DNA replication. The mismatch errors are recognized by the MSH2/MSH6 and MSH2/MSH3 heterodimeric complexes and repaired by the MLH1/PMS2 heterodimers. In the case of MMR deficiency (dMMR), mutations accumulate throughout the genome, particularly in microsatellite regions where the length of mononucleotide and dinucleotide repetitions is altered. The presence of MSI can be detected by a polymerase chain reaction (PCR), amplifying specific microsatellite loci [64], or indirectly by immunohistochemistry (IHC), assessing the nuclear expression of MLH1, MSH2, MSH6, PMS2 [65]. In the case of PCR, the presence of MSI is usually referred to as MSI-high (MSI-H), while its absence is indicated as MSI-low (MSI-L) or microsatellite stability (MSS). In the case of IHC, the lack of expression of one of MMR-related proteins is usually referred to as dMMR, while their normal expression is indicated as MMR proficiency (pMMR).

Several cohort analyses in different types of cancer, including gastric cancer, have provided consistent evidence that MSI-H/dMMR is associated with better long-term outcome, resistance to chemotherapy, and responsiveness to ICI [66].

A post hoc analysis of the MAGIC trial was carried out to correlate MSI with survival and response to perioperative chemotherapy. Among patients who received surgery alone, those with MSI-H/dMMR tumors showed a better survival as compared to those with MSS/pMMR (HR 0.42; 95% CI, 0.15–1.15; *p* = 0.09). In contrast, among patients treated with chemotherapy plus surgery, those with MSI-H/dMMR tumors had a worse prognosis, with a median OS of 9.6 months vs. 19.5 months in the MSS/pMMR group (HR 2.18, 95% CI, 1.08–4.42; *p* = 0.03) [67]. Similarly, a post hoc analysis of the CLASSIC trial showed that adjuvant chemotherapy improved DFS in MSS group (5-year DFS rate 66.8% vs. 54.1%; *p* = 0.002), but not in the MSI-H one (5-year DFS rate 83.9% vs. 85.7%; *p* = 0.931) [68].

The favorable prognosis and the lack of benefit from chemotherapy observed in patients with MSI-H/dMMR resectable gastric cancer has been confirmed in a recent meta-analysis [69], and will likely change the decision making process in patients with localized disease.

The high anti-tumor activity of ICI in the presence of MSI has been well established in different types of metastatic tumor [70,71], leading the US Food and Drug Administration (FDA) to approve Pembrolizumab, an anti-PD1 monoclonal antibody, in all patients with MSI-H metastatic tumors, independently from histology.

The efficacy of ICI was shown also in MSI-H metastatic gastric cancer [72], but presently no data are available from phase III clinical trials in patients with resectable gastric cancer. Encouraging results have been observed in two recent phase II studies, the GERCOR NEONIPIGA trial [73] and the DANTE trial [74]. The first study evaluated the efficacy of neoadjuvant nivolumab and ipilimumab, followed by adjuvant nivolumab, in patients with resectable MSI/dMMR, T2-T4 Nx M0 gastric cancer. All of the 29 patients who underwent surgery had R0 resection and 17 (59%) had pathological complete response (pCR) (i.e., ypT0N0) [73]. The second study evaluated atezolizumab in the perioperative treatment of resectable gastric or gastroesophageal junction (GEJ) cancer in combination with FLOT chemotherapy. A higher rate of pCR was observed in the atezolizumab group compared to the control group (50% vs. 27%) [74].

Intriguingly, according to the TCGA molecular classification [57], the prevalence of MSI is higher in patients with resectable gastric cancer (20%) and very low in those with metastatic disease (<5%). As a consequence, patients with resectable tumors could benefit the most from ICI treatment, and several clinical studies are investigating the possible contribution of these agents in the perioperative or adjuvant treatment (Table 3).

It would not be surprising if these trials duplicate what has been recently observed in colorectal cancer [75]. Given that MMRd metastatic colorectal cancer is responsive to ICI, it has been hypothesized that patients with MMRd and locally advanced rectal cancer could benefit as well. Thus, a prospective phase 2 study was started in which patients with MMRd stage II/III rectal cancer received dostarlimab, an anti-PD-1 monoclonal antibody. Twelve consecutive patients enrolled in the study experienced a clinically complete response as evidenced by magnetic resonance imaging and endoscopic biopsy, and no cases of progression or recurrence had been reported during follow-up (range 6–25 months) [75]. Thus, localized MMRd rectal tumors seem to respond to ICI even better than metastatic colorectal tumors. We have to wait at least until interim analyses of the ongoing clinical studies are available to verify whether the same can happen for gastric cancer.

### 5.3. PD-L1 Expression

PD-L1 is a transmembrane glycoprotein expressed in cancer cells and in tumor-infiltrating immune cells, mostly lymphocytes and macrophages. By binding PD-1, a transmembrane protein expressed in T-cells, PD-L1 determines the inhibition of innate and adaptative immune response [76]. Therefore, the PD-1/PD-L1 pathway represents a tumor escape mechanism in response to endogenous anti-tumor activity. The inhibition of PD-1/PD-L1 interactions, through anti-PD1 or anti-PD-L1 monoclonal antibodies, is able to reactivate immune response against cancer and has profoundly changed the treatment of several tumors, including gastric cancer, especially in the metastatic phase [77].

PD-L1 expression is evaluated by IHC based on the combined positive score (CPS), i.e., the number of PD-L1-positive cells (tumor, lymphocytes and macrophages) in relation to total viable tumor cells. In some studies, even the tumor proportion score (TPS), i.e., the number of positive tumor cells in relation to all tumor cells, has been utilized.

As observed in other tumors, higher PD-L1 expression is associated with better response to ICI in advanced gastric cancer. A CPS ≥ 1 has been associated with response to Pembrolizumab [78], but a CPS ≥ 10 was associated with a more considerable therapeutic benefit in a study in which Pembrolizumab was compared to Paclitaxel as third-line treatment [79]. In a similar study, Nivolumab also improved OS, apparently regardless of PD-L1 expression, but in this case PD-L1 was determined by TPS [80]. In a subsequent trial carried out on newly diagnosed patients with metastatic gastric cancer, Nivolumab plus chemotherapy in a first-line setting showed a significant improvement of OS, as compared to chemotherapy alone in patients with PD-L1-positive tumor CPS ≥ 1. However, a greater benefit was observed in PD-L1 CPS ≥ 5, indicating that Nivolumab provides superior efficacy in patients with higher PD-L1 CPS [81].

As proposed by the TCGA molecular classification [57], EBV-positive and MSI subtypes frequently express high levels of PD-L1; this could explain, at least in part, the intense immune responsiveness of these tumors. Interestingly, in a phase 2 trial in which 61 unselected patients with metastatic gastric cancer were treated with Pembrolizumab as a salvage therapy, MSI-H and EBV positivity were mutually exclusive conditions and, in these patients, the absence of expression of PD-L1 was associated with resistance to Pembrolizumab (no patient responded to the treatment), while the ORR was 50% in patients with PD-L1 CPS ≥ 1 [60]. In another retrospective analysis, 259 unselected patients with metastatic gastric cancer and treated with Pembrolizumab after failure of two previous lines of chemotherapy showed an ORR of 16.4%, but when the analysis was stratified according to PD-L1 expression, patients who were PD-L1 negative (N = 109) had an ORR of only 8.6%, while those with PD-L1 CPS ≥ 1 (N = 149) had an ORR of 22.7% [78]. Moreover, when considering only patients with MSS, i.e., after excluding MSI-H tumors, the ORR decreased to 13.3% in PD-L1-positive tumor and even more, 9%, in those PD-L1-negative [78]. These data indicate that MSI-H and PD-L1 can both predict response to ICI, but their co-occurrence in tumor cells could more accurately do so.

The above-reported data suggest that PD-L1 expression, assessed according to CPS, along with MSI, should be considered in future clinical trials investigating the role of ICI in the adjuvant/perioperative setting of gastric cancer.

### 5.4. Epstein–Barr Virus (EBV)

EBV, known to cause Burkitt lymphomas and undifferentiated nasopharyngeal carcinomas, was more recently found to be associated with about 10% of gastric cancer [57]. The detection of EBV-encoded small RNA (EBER) by in situ hybridization (EBER-ISH) is the standard technique for the evaluation of EBV-infected cells in tissue samples [82]. EBV-positive gastric cancers exhibit abundant T-cell infiltrates in the tumor microenvironment and high PD-L1 expression [59] and, as a consequence, are very responsive to ICI [60]. In a small cohort of 61 patients with gastric cancer treated with Pembrolizumab, all patients with EBV-positive tumors (N = 6) responded to the treatment (ORR 100%). As expected, ORR was high also in patients with PD-L1-positive tumors (N = 28) or MSI-H tumors (N = 7), 50% and 85.7%, respectively [60].

Although EBV status is not currently recommended for routine clinical care, the presence of EBV, likewise MSI and PD-L1 expression, should be actively considered in clinical trials investigating ICI activity in gastric cancer.

### 5.5. Specific Molecular Targets

At present, HER2 is the only molecular target that guides therapeutic decisions in advanced gastric cancer. In patients with metastatic HER2-positive tumor, the addition of trastuzumab (anti-HER2 monoclonal antibody) to platinum-based chemotherapy has been proven to increase median OS compared to chemotherapy alone [83]. The positivity of HER-2 is assessed by IHC or ISH for gene amplification in the case of equivocal results.

Very few data have been provided on the efficacy of Trastuzumab in the perioperative/adjuvant setting. A retrospective cohort study compared chemotherapy with or without Trastuzumab among patients with HER2-positive locally advanced gastric cancer in the adjuvant or neoadjuvant settings [84]. Trastuzumab was shown to improve ORR, but not OS. Given the observed limited clinical benefit, ongoing clinical trials are designed with experimental arms in which the antitumor effect of Trastuzumab is potentiated by the addition of Pertuzumab (another anti-HER2 monoclonal antibody; EORTC-Innovation trial), or Atezolizumab (anti-PD-L1; NCT04661150 trial) (Table 3).

Other targeted therapies have been investigated in clinical trials in patients with advanced gastric cancer, including Cetuximab and Panitumumab in EGFR-positive tumor, or Rilotumumab and Onartuzumab in MET-positive tumor [85]. No benefit in outcomes has been observed by adding these targeted agents to the standard chemotherapy. In tumor with FGFR2 amplification, trials on the efficacy of inhibitors of FGFR2, such as Dovitinib (NCT01719549) or AZD4547 (NCT01457846), are ongoing in the metastatic setting and, therefore, are far away to be tested in resectable gastric cancer.

## 6. Conclusions

Despite advancement in surgical technique, including D2 dissection and retrieval of more than 16 lymph nodes, and the clinical benefit obtained with novel chemotherapeutic regimens (such as FLOT), long-term outcome of patients affected by gastric cancer is still disappointing. Currently, three different approaches, i.e., adjuvant chemotherapy, adjuvant chemoradiotherapy, and perioperative chemotherapy, have been demonstrated to improve survival over surgery alone. We believe that an international phase III clinical trial should be performed in order to establish if a strategy is superior to the others. A more accurate definition of patients potentially gaining a benefit from chemotherapy, along with novel biomarkers, such as MSI and EBV, able to identify patients who are most likely to benefit from immune therapies, might improve outcomes in the near future. At present, limited data are available for EBV-positive tumors and most of the studies that have established the benefit of ICI in patients with MSI-H tumors have been performed in patients with advanced disease; therefore, ad hoc trials are required to translate these results in patients with resectable gastric cancer. Ongoing phase III clinical trials are evaluating if the administration of immune therapy in MSI-H gastric cancer in the adjuvant/perioperative setting can produce a similar survival benefit as that observed in the advanced disease.

## Figures and Tables

**Table 1 cancers-14-04670-t001:** Phase III clinical trials guiding the current clinical practice.

Study (Year)/Region	Treatment Arms	No of Patients	Outcome	HR (95% CI)	*p **
** ADJUVANT CT **					
ACTS-GC (2007)/ Japan	Surgery (D2) alone Surgery (D2) + S-1	530 529	5-yr-OS 61% 5-yr-OS 72%	1 0.67 (0.54–0.83)	**0.003**
JACCRO-GC-07 (2022)/Japan	Surgery (D2) + S-1 Surgery (D2) + S-1 + DTX	459 454	3-yr-DFS 50% 3-yr-DFS 66%	1 0.632 (0.40–0.99)	**0.001**
CLASSIC (2012)/ South Korea	Surgery (D2) alone Surgery (D2) + CAPOX	515 520	5-yr-OS 69% 5-yr-OS 78%	1 0.66 (0.51–0.85)	**0.001**
** ADJUVANT CT + RT (CHEMORADIOTHERAPY) **					
INT-0116 (2001)/ North America	Surgery alone Surgery + 5FU/FA/RT	275 281	3-yr-DFS 41% 3-yr-DFS 50%	1.35 (1.09–1.66) 1	**0.005**
ARTIST (2012)/ South Korea	Surgery (D2) + XP Surgery (D2) + XP/RT	228 230	5-yr-OS 73% 5-yr-OS 75%	1.13 (0.78–1.65) 1	0.530
ARTIST-II (2021)/ South Korea	Surgery (D2) + S-1 in N+ Surgery (D2) + SOx in N+ Surgery (D2) + SOx/RT in N+	182 181 183	3-yr-DFS 65% 3-yr-DFS 74% 3-yr-DFS 73%	1.44 (1.02–2.44) 1 1.10 (0.85–1.61)	**0.042** 0.879
** ADJUVANT CT AFTER NEOADJUVANT CT AND SURGERY (PERIOPERATIVE CT) **					
MAGIC (2006) Europe	Surgery alone ECF + Surgery + ECF	253 250	5-yr-OS 23% 5-yr-OS 36%	1 0.75 (0.60–0.93)	**0.009**
AIO-FLOT4 (2017) Germany	ECF(X) + Surgery + ECF(X) FLOT + Surgery + FLOT	360 256	5-yr-OS 36% 5-yr-OS 45%	1 0.77 (0.63–0.94)	**0.012**

* Significant values (*p* < 0.05) are in bold. CT, chemotherapy; RT, radiotherapy; DTX, docetaxel; CAPOX, capecitabine plus oxaliplatin; 5FU, 5-fluorouracil; FA, folinic acid; XP, capecitabine plus cisplatin; SOx, S-1 plus oxaliplatin; ECF, epirubicin plus cisplatin plus 5-fluorouracil; (X), capecitabine instead of 5-fluorouracil; FLOT, 5-fluorouracil plus folinic acid plus oxaliplatin plus docetaxel.

**Table 2 cancers-14-04670-t002:** Number of patients at stage IB enrolled in clinical trials.

Trial	Included Stages	N. of Patients at Stage IB (%)	Staging System
ACTS-GC	II-IIIB	0	AJCC 2nd edition
JACCRO-GC-07	III	0	AJCC 6th edition
CLASSIC	II-IIIB	0	AJCC 6th edition
INT-0166	IB-IV (M0)	62 (11.2)	AJCC 3rd edition
ARTIST	II-IIIB	99 (21.6)	AJCC 6th edition
MAGIC	II-IV (M0)	0	AJCC 5th edition
FLOT	II-IV (M0)	113 (27.2)	AJCC 7th edition

**Table 3 cancers-14-04670-t003:** A selection of active clinical trials in the adjuvant/perioperative treatment of resectable gastric cancer.

Study	Phase	Setting	Arms	Eligibility	Endpoint
** CONVENTIONAL CHEMOTHERAPY **					
NCT01787539 stoperopchem	II/III	Perioperative	EOX-Surgery-EOX EOX-Surgery-Observation	cT2-4a, N0-3	DFS OS
** IMMUNE CHECKPOIN INHIBITORS (ICI) * **					
NCT04744649	II	Neoadjuvant	CAPOX or SOX CAPOX or SOX + **JS001**	PD-L1 ≥ 10% or EPV+ or MSI-H/dMMR	pCR
NCT04795661 Imhotep	II	Neoadjuvant	**Pembrolizumab**	EPV+ or MSI-H/dMMR	pCR
NCT04139135	III	Perioperative	SOx SOx + **HLX10**	PD-L1 ≥ 5%	EFS
NCT03221426	III	Perioperative	XP(F) or FLOT XP(F) or FLOT + **Pembrolizumab**	≥cT3 or cN+	pCR OS EFS
NCT04592913	III	Perioperative	FLOT FLOT + **Durvalumab**	Stage III	EFS
NCT03006705 Attraction-5	III	Adjuvant	CAPOX or S1 CAPOX or S1 + **Nivolumab**	Stage III and Surgery R0	RFS
** TARGETED AGENTS ± ICI * **					
EORTC- Innovation	II	Perioperative	CT CT + Trastuzumab CT + Trastuzumab + Pertuzumab	HER2-positive	pCR
NCT04661150	II	Perioperative	CAPOX + Trastuzumab CAPOX + Trastuzumab + **Atezolizumab**		pCR

* ICI agents are in bold. CT, chemotherapy; EOX, epirubicin plus oxaliplatin plus capecitabine; CAPOX, capecitabine plus oxaliplatin; SOx, S-1 plus oxaliplatin; XP, capecitabine plus cisplatin; (F), 5-fluorouracil instead of capecitabine; FLOT, 5-fluorouracil plus folinic acid plus oxaliplatin plus docetaxel; DFS, disease-free survival; OS, overall survival; pCR, pathological complete response; EFS, event-free survival, RFP, relapse-free survival.

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
