# Peer review of "Optimizing the Choice for Adjuvant Chemotherapy in Gastric Cancer"

_cancers, 2022, doi:10.3390/cancers14194670_

Round 1
Reviewer 1 Report (Previous Reviewer 2)
The responses are satisfactory
Reviewer 2 Report (Previous Reviewer 1)
The authors addressed all the comments. The manuscript deserves to be accepted.
Reviewer 3 Report (Previous Reviewer 4)
I think the manuscript has been sufficiently improved for publication in the Cancers.
This manuscript is a resubmission of an earlier submission. The following is a list of the peer review reports and author responses from that submission.
Round 1
Reviewer 1 Report
Dr. Grassadonia and colleagues presented an excellent overview of the current trend in gastric cancer treatment. They focused on the optimal choice of adjuvant chemotherapy based on the different regional strategies. They also provided a comprehensive review of the pathologic features that should be identified for high risk of recurrence patients. Finally, they reviewed the current advances in biomarkers and therapeutics with potential for personalized best management.
The authors should be congratulated for their excellent work and thorough analysis of an important topic. However, their results should be slightly revised to give them more precision to support their conclusions. Some comments for further improvement of the manuscript include:
Major
Conclusion:
Despite the argument and data provided on the promising results of treatment of advanced MSI-h colorectal cancer, the authors should moderate this sentence with the use of the conditional. In addition, this sentence tends to be confusing by bringing colorectal cancer into the conclusion. The authors could simply reword their thinking to be more specific and not leave room for confusion. Finally, the authors could have emphasized the need for an international trial comparing the three regimens currently used in different regions.
Minor
-
Section 2.2: The authors could provide more precision on the HR results, giving the results in terms of 95% CIs. This would give the reader a clue about dispersion, sample size, etc.
-
Section 2.3: the authors should also mention modified D1.5 or D2 lymphadenectomy.
-
Section 2.4: authors should pay more attention to precision. They could consider either giving specific RFS results in a summary table or including them with a 95% CI. Same comment for risk of recurrence.
-
Section 2.5: same comment as in section 2.4. Authors could give more precision, for OS and RFS 161 lines.
Again, I enjoyed reading this excellent and important manuscript!
Reviewer 2 Report
The authors have performed a comprehensive review and identified various prognostic and predictive pathological and molecular markers but it is not clear from the paper how to optimise the choice of chemo. There are challenges when using Asian and non-Asian studies given that the outcomes in non-metastatic disease is very different following multimodality treatment as evidence by the much higher 5 yr OS in Asian cohorts. There are are a few areas
1) Stage 1ds - as alluded to, the std of care is observation alone. Though Stage IB cancers may sometimes have a prognosis similar to Stage 2, in the absence of randomised data, routine adjuvant chemo cannot be recommended. Also, pT2N0 is not the same as cT2N0 when trying to select which is the better approach which the author did not differentiate in section 4.2. Current imaging cannot reliably stage gastric cancer, hence cT2N0 tumour could be more advanced than imaged and therefore peri-op chemo can be considered.
2) The authors do not bring to attention that in Stage 3 or node+ ds, a doublet adjuvant therapy is preferred over S-1 monotherapy as evidenced by JACCRO GC-07 study and the ARTIST-2 study with better survival.
3) The RESOLVE study Lancet Oncology 2021 22:1081 demonstrated that in cT4 ds, periop SOX improved RFS over adjuvant CAPOX. I would suggest that this study is referenced.
4) The apparent lack of benefit of chemo in MSI-H/dMMR tumours from CLASSIC and MAGIC trials is suggestive but currently not practise changing as they are retrospective in nature and small numbers of patients. Ref 50 which the authors cited, suggest there may still be benefit of adjuvant chemo in MSI-H tumours. Recent abstracts of prospective trials DANTE and NEONIPIGA suggest benefit of adding immune checkpoint inhibitors in the pre-op therapy, consider referencing these.
Reviewer 3 Report
This manuscript has been described current status and future direction of the adjuvant chemotherapy in gastric cancer as review.
In section 4.2 (Stage I), author described it is not necessary to perform adjuvant chemotherapy for the patients in pT1N1 treated with appropriate lymph node dissection. This content was endlessly repetitive, so it should be more concise.
Line 451-452 is also unnecessary.
In Table 2, CLASIC Trial was described duplicated.
In section 5.2, author introduced the clinical trial for the MMRd patients with colorectal cancer. In section 5.1, author already described ICI might be effective for the patients with MSI-high as adjuvant setting, so line 472 to 482 is unnecessary.
Line 461 and 462: patients with early gastric cancer or patients with early stage is incorrect, patients with resectable gastric cancer is correct representation.
Section 5.3 was described similar contents with section 5.1.
It has been reported that skeletal muscle mass, nutritional status, and immunological background are involved in the completion of treatment, and should be mentioned in addition to tumor characteristics.
Reviewer 4 Report
In this article, the authors have reviewed the clinical trials that guide the current clinical application of adjuvant treatment for patients affected by resectable gastric cancer. They have focused on different approached including adjuvant chemotherapy, adjuvant chemoradiotherapy, and perioperative chemotherapy. The manuscript is well-written and sufficiently support the topic. I would recommend this manuscript for publication after considering few revisions:
- It would be better if the authors could provide a brief background and description of adjuvant chemotherapy, adjuvant chemoradiotherapy, and perioperative chemotherapy, preferably in the introduction part. This will help non-clinician scientists to follow the paper.
- A summarizing table for description of limitations/opportunities of three adjuvant-based methods with focus on the discussed clinical trials would be helpful.
- If applicable, please provide name/information about the adjuvant type when discussing about adjuvant-based clinical trials, for example in the table.